# Amine-Based Deep Eutectic Solvents for Alizarin Extraction from Aqueous Media

**Nihal Yasir [1], Amir Sada Khan [1,2], Noor Akbar [3], Muhammad Faheem Hassan [1], Taleb H. Ibrahim [1,*], Mustafa Khamis [3] , Ruqaiyyah Siddiqui [3] , Naveed Ahmed Khan [4] and Paul Nancarrow [1]**

1   Department of Chemical Engineering, College of Engineering, American University of Sharjah, Sharjah P.O. Box 26666, United Arab Emirates; g00087576@aus.edu (N.Y.); aamirsada_khan@yahoo.com (A.S.K.); mhassan@aus.edu (M.F.H.); pnancarrow@aus.edu (P.N.)
2   Department of Chemistry, University of Science & Technology, Banuu 28100, Pakistan
3   Department of Biology, Chemistry and Environmental Sciences, American University of Sharjah, Sharjah P.O. Box 26666, United Arab Emirates; noormicrobiologist555@gmail.com (N.A.); mkhamis@aus.edu (M.K.); rsiddiqui@aus.edu (R.S.)
4   Department of Clinical Sciences, College of Medicine, University of Sharjah, Sharjah P.O. Box 26666, United Arab Emirates; naveed5438@gmail.com
*   Correspondence: italeb@aus.edu

**Abstract:** Alizarin dye is toxic and has a negative influence on human life and the environment. Consequently, the scientific community faces a difficult issue in developing efficient approaches for removing alizarin from water streams. Six distinct deep eutectic solvents (DESs) containing different hydrogen bond acceptors (HBAs), namely trioctylphosphine, trioctylamine and trihexylamine, and two hydrogen bond donors (HBDs), namely salicylic acid and malonic acid, were used to rapidly remove alizarin from high concentration solutions up to 2000 mg/L at room temperature using the liquid–liquid micro-extraction method (LLE). DES-3 had the highest extraction efficiency for alizarin among the other synthesized DESs. The effect of process variables such pH, contact time, dye initial concentration, volume ratio, temperature and salt on alizarin extraction efficiency from water stream was explored, optimized and reported. Statistical analysis was conducted to ensure the accuracy of values for the optimized parameters. For a 1000 mg/L solution of alizarin with a DES/alizarin volume ratio of 1:10 at room temperature, the maximum elimination of 98.02 percent was achieved in 5 min. FTIR was used to analyze the structural properties of DES and the interaction between DES and alizarin. The thermal stability of DES-3 was determined using thermogravimetric analysis (TGA) and indicated that DES-3 has excellent thermal stability up to 320 °C. Human saline was used to test the toxicity of the synthesized DES in vitro. It was determined that synthesized DES is less harmful and more effective at removing alizarin.

**Keywords:** deep eutectic solvents; alizarin; liquid–liquid extraction

## 1. Introduction

Several dyes and pigments, some of which are poisonous or carcinogenic, are regularly released into the environment through industrial wastewater discharge [1]; consequently, these effluents can have a significant impact on the aquatic ecosystems [2–4]. The dye, food coloring, cosmetics, paper, and textile industries are responsible for the majority of these contaminations [5–7]. The presence of these dyes in wastewater can cause both pollution and harm to the environment [8]. Alizarin Red S (ARS) is an anthraquinone dye that belongs to the category of dyes with the highest resistance. It has been used in the textile business for so long that its release into the environment is considered as a significant threat to humans and animals [9].

Purification of the discharged wastewater containing these compounds is difficult due to the dye molecules' resistance to treatment by conventional methods [8]. Liquid membrane [10], membrane filtration [11], liquid–liquid extraction (LLE) [12], adsorption [13],

coagulation [14], ion exchange [15], and photocatalytic degradation [16] are some of the available treatment options. Some of these approaches, however, have shown disadvantages such as high cost, long duration, considerable labor, and limited removal efficiency. Solvent extraction is a common treatment method used in the petroleum, chemical and food sectors because of its decreased energy costs. On the other hand, LLE has the ability to remove a dye efficiently and swiftly with only a tiny amount of solvent. Ionic liquids (ILs) and deep eutectic solvents (DESs) are two of the most prominent solvents utilized in dye LLE procedures [17].

ILs are ionic solvents having excellent thermal and chemical stability, low vapor pressures, good water miscibility and high melting temperatures. By combining certain anions and cations, ILs with desired characteristics may be produced. These 'tunable' ILs have the ability to effectively remove dyes from the aqueous phase by LLE [18,19]. Most hydrophobic ILs, on the other hand, contain fluorine and can generate poisonous HF when hydrolyzed [20]. In addition, ILs are more difficult to synthesize and more costly than DESs. Their usage as alizarin extractors is limited due to their toxicity and expensive cost. In LLE applications, researchers have been looking for alternatives to hydrophobic ILs.

DESs, which are essentially modified ILs, have lately emerged as possible LLE solvents for a variety of contaminants, including alizarin. They are made up of a hydrogen bond donor (HBD) and a hydrogen bond acceptor (HBA) that are combined together in precise molar ratios. These DESs are more cost-effective and environmentally friendly when compared with ILs. They are better than ILs in terms of cost, more environmentally friendly, easy to manufacture, have equivalent or better physiochemical characteristics, and are rich in hydrogen and ionic bonding [21–23]. Low melting points and vapor pressures, in addition to excellent solubility, biodegradability, and thermal and chemical stability, are all common characteristics of DESs [23–25]. These characteristics explain why DESs have recently become more widely used in a variety of sectors [26]. Because of their immiscibility in water, hydrophobic DESs are often utilized in LLE applications. They have recently been utilized to extract different metal ions and dyes from aqueous solutions [27,28]. Their potential uses as a solvent or in surface modification for dye removal through LLE have also been investigated. For the extraction of Eriochrome black T, Kaur et al. employed glycolic acid and choline chloride-based DES [17].

In the present work, six hydrophobic DESs were synthesized utilizing three HBAs, namely trioctylphosphine (TOP), trioctylamine (TOA), and trihexylamine (THA), in addition to two HBDs, namely salicylic acid (SA) and malonic acid (MA). Consequently, TOP:SA, TOP:MA, TOA:SA, TOA:MA, THA:SA and THA:MA are represented, respectively, by DES-1, DES-2, DES-3, DES-4, DES-5 and DES-6. DES-3 was shown to have the best efficiency for alizarin extraction from the aqueous phase among the six DESs tested. As a result, it was chosen for further optimization research. Effects of pH, contact duration, initial alizarin concentration, temperature, ionic strength and DES on alizarin volume ratio ($V_{DES}$:$V_A$) were all optimized throughout the LLE process. The interaction between DES and alizarin was studied using the FTIR characterization technique. The thermodynamic characteristics were used to examine the feasibility of alizarin interaction from aqueous media to DES's phase. Human cells were used to test the toxicity of DES-3 in vitro. Statistical analysis of the published articles on dyes extraction from year 2016 to 2022 is shown in Figure 1.

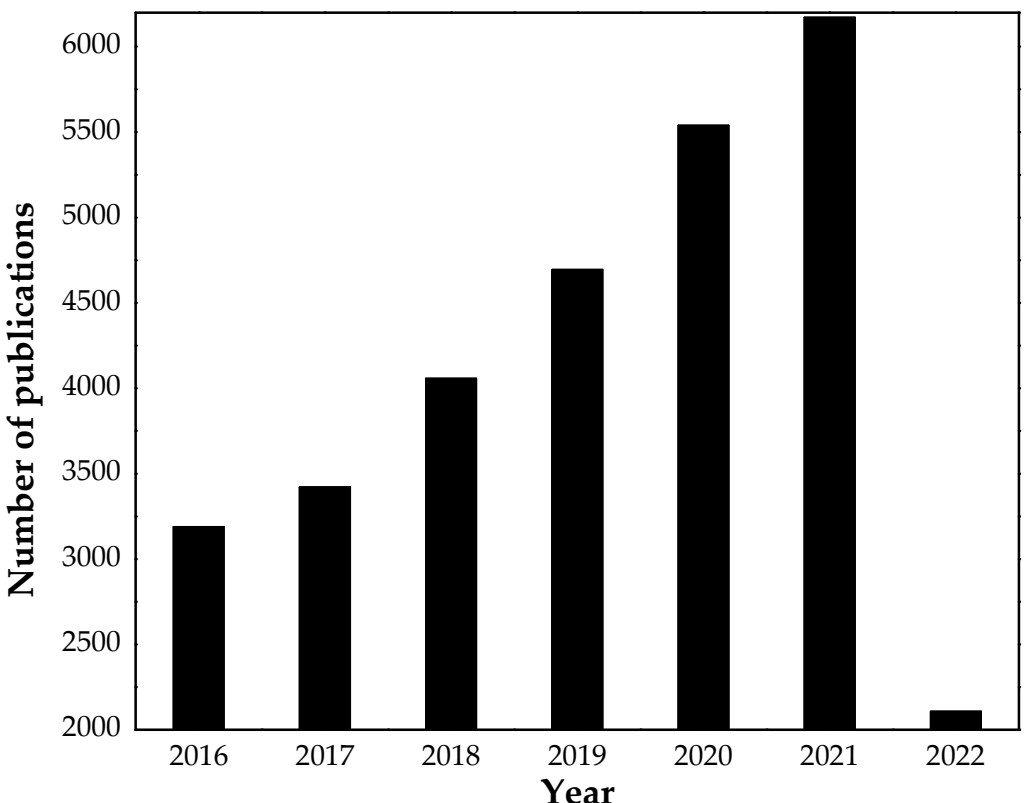

**Figure 1.** Total number of articles in Scopus about dye removal by liquid–liquid extraction from 2016 to 2022.

## 2. Experimental

### 2.1. Materials

All primary chemicals were purchased from Sigma Aldrich, of analytical reagent grade, and used as received without any further purification unless otherwise specified. Trioctylamine ($C_{24}H_{51}N$, 98%), trioctylphosphine ($C_{24}H_{51}P$, 97%), trihexylamine ($C_{18}H_{39}N$, 96%), salicylic acid (2-hydroxy benzoic acid) ($C_7H_6O_3$, 99%) and malonic acid ($C_3H_4O_4$, 99%) were all used in the preparation of the different DESs. Analytic grade Alizarin Red S ($C_{14}H_7NaO_7S$) was purchased from Selleck Chemicals, USA, having dye content of about 99%. Alizarin stock solution (2000 mg/L) was prepared by dissolving 2 g of alizarin in 1 L of distilled water. Furthermore, the working solutions of different concentrations were prepared by diluting the parent stock solution. Sodium hydroxide (NaOH, 99%) and hydrochloric acid (HCl, 37%) were both procured from Merck, Germany.

### 2.2. Preparation of the DESs

A molar ratio of 1:1 of salicylic acid and trioctylamine was used to synthesize the DES used in this study (DES-3). DES-3 was prepared by adding 0.69 g (1 mol) of salicylic acid (HBD) into the heated 1.77 g (1 mol) of trioctylamine (HBA) in a 20 mL scintillation vial at 60 °C. Stirring using a heat-stirrer (Stuart CB162, UK) continued for 1 h to get a homogenous clear DES and then left to cool at room temperature. Other DESs were also prepared using the same above-mentioned procedure by adding an HBD to a heated HBA. Materials and amounts of the other prepared DESs are shown in Table 1. Figures 2 and 3 display the schematic diagram and the physical appearance of the prepared DES in this work, respectively. The chemical structures of HBA and HBD precursors for the DES prepared in this work are shown in Figure 4.

**Table 1.** HBDs and HBAs and their proportion used for the preparation of DESs.

| Name | Materials | Amounts (g) | Molar Ratio |
|---|---|---|---|
| DES-1 | Salicylic acid | 0.69 | 1:1 |
| | Trioctylphosphine | 1.85 | |
| DES-2 | Salicylic acid | 0.69 | 1:1 |
| | Trihexylamine | 1.01 | |
| DES-3 | Salicylic acid | 0.69 | 1:1 |
| | Trioctylamine | 1.77 | |
| DES-4 | Malonic acid | 0.52 | 1:1 |
| | Trioctylphosphine | 1.85 | |
| DES-5 | Malonic acid | 1.04 | 1:1 |
| | Trihexylamine | 1.01 | |
| DES-6 | Malonic acid | 0.52 | 1:1 |
| | Trioctylamine | 1.77 | |

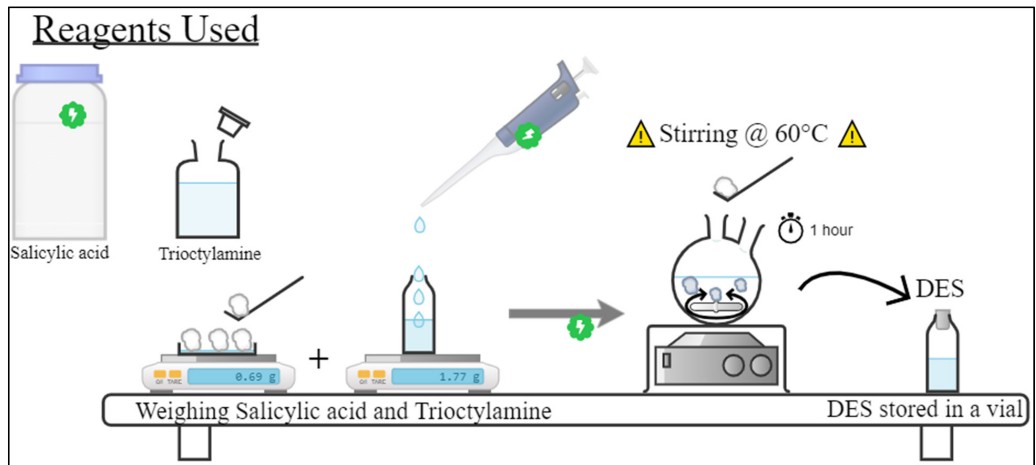

**Figure 2.** Schematic diagram for the preparation of the DESs.

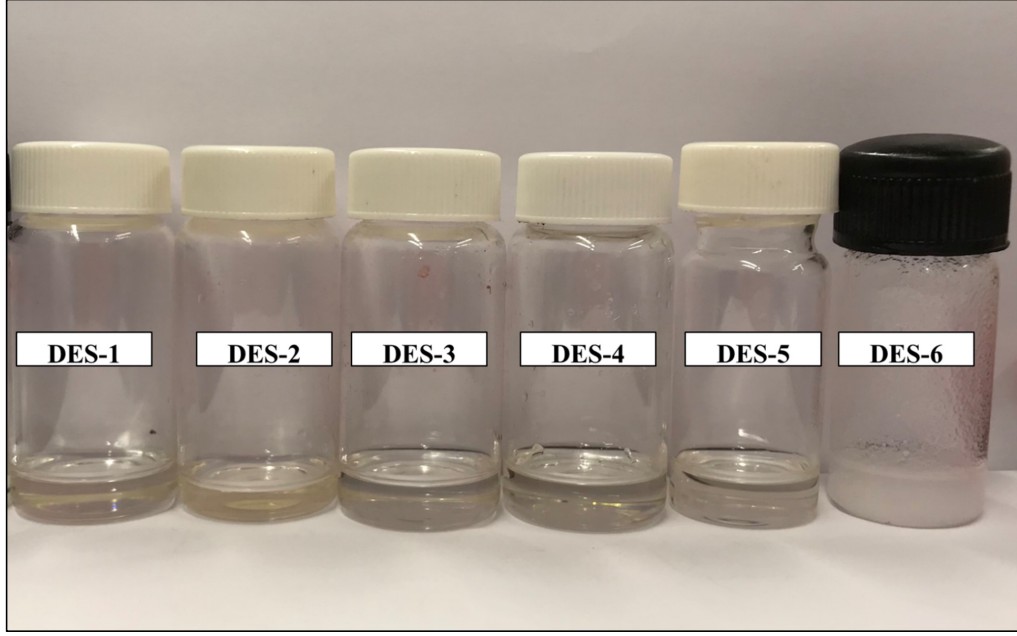

**Figure 3.** The prepared DESs.

**Figure 4.** HBA and HBD making up the different DESs.

## 3. Characterization and Instruments

### 3.1. Instruments

In this study, many sorts of analytical methods were used. Using an Oakton pH meter 510 series (Malaysia), the pH of all the solutions was determined. A vortex mixer was used to mix both the DES and alizarin solution together (Stuart, UK). A centrifuge was used to separate both DES and alizarin aqueous solutions (HERMLE Labortechnik, Germany). The alizarin concentration in the aqueous phase following DES extraction was measured using a Thermo Scientific Evolution 220 UV-Vis spectrophotometer produced in China. The materials were analyzed using a Perkin Elmer Fourier transform infrared spectrophotometer (FTIR) (Thermo Scientific, Waltham, MA, USA). The pH of the alizarin solution was measured by a pH meter (Oakton pH 510 series, Malaysia). Distilled water was generated using an Aquatron A4000D Water Still, UK.

### 3.2. Liquid–Liquid Extraction Study

The extraction of alizarin using different DESs from aqueous media was investigated. In a series of 10 mL culture tubes containing 1.5 mL alizarin dye solution of known concentrations (200 mg/L) and pH (6.37), a sufficient dose of the prepared DES (0.15 mL) was applied, and the culture tube was immediately placed in a vortex mixer (Stuart, UK) for 5 min. A cloudy solution emerged, and a fine layer of dye was extracted from the initial dye solution. The mixture was then centrifuged for 10 min at 2500 rpm. DES, along with the extracted dye, were found to be settled at the top of the centrifuge culture tube, leaving clear liquid behind. Thus, two phases of DES-extracted dye and a clear aqueous solution

were clearly visible. Subsequently, samples of the bottom liquid were analyzed for dye concentration using the UV-vis spectrophotometer at $\lambda_{max} = 261$ nm. The percent extraction efficiency (%E) and the experimental partition coefficient (*p*) of alizarin dye in DES were calculated using the following equations:

$$\%E = \frac{C_i - C_f}{C_i} \times 100 \tag{1}$$

$$p = \frac{C_i - C_f}{C_f}\left(\frac{V_{aq}}{V_{DES}}\right) \tag{2}$$

where $C_i$ and $C_f$ represent the initial and final dye concentration, and $V_{aq}$ and $V_{DES}$ are the volume of aqueous solution and volume of DES, respectively.

### 3.3. Single Parameter Optimization Process

Various process factors, such as starting pH, vortex mixing duration, DES to alizarin solution volumetric ratio, initial concentration of alizarin solution, temperature, and anionic salt, have a significant impact on alizarin extraction from liquid phase to DES's phase. The adjustment of these process parameters is critical to achieve the maximal removal of alizarin from the aqueous phase. Individual parameter optimization is covered in detail in the subsections that follow. A brief summary of the optimization process is illustrated in Table 2.

**Table 2.** Summary of the optimization process for the extraction of alizarin dye by DES-3.

| Studied Parameter | Value | Fixed Parameters | | | | |
| --- | --- | --- | --- | --- | --- | --- |
| | | pH | Time (mins) | $C_i$ (mg/L) | DES Dosage (mL) | Temperature (°C) |
| pH | 2–12 | - | 5 | 200 | 0.15 | 25 |
| Time | 0.5–10 min | 6.37 | - | 1000 | 0.15 | 25 |
| Initial concentration | 200–2000 mg/L | 6.37 | 5 | - | 0.15 | 25 |
| Volumetric ratio | 1:10 to 1:60 | 6.37 | 5 | 1000 | - | 25 |
| Temperature | 25–60 °C | 6.37 | 5 | 1000 | 0.15 | - |

#### 3.3.1. Effect of Initial pH of Solution

According to earlier studies, the initial pH of alizarin solution has a significant impact on its extraction efficiency. Changing the ionization degree of alizarin molecules has an influence on the extraction process [29]. It is one of the most critical factors, and adjusting the pH of the alizarin solution is necessary before optimizing other process parameters. pH optimization tests were carried out using alizarin solution at a concentration of 200 mg/L over a pH range of 2–12. The experiment was carried out at room temperature with 5 min of mixing and a $V_{DES}$:$V_w$ ratio of 1:10 mL. Analytical grade 0.1 N HCl and 1 N NaOH solutions were used to alter the pH of the alizarin solution at the start.

#### 3.3.2. Effect of Time

The tests were carried out in a time range of 0.5 to 10 min, mixing 1000 mg/L of alizarin with $V_{DES}$:$V_w$ in a 1:10 volumetric ratio at room temperature to examine the influence of vortex mixing duration.

#### 3.3.3. Effect of Initial Concentration of Alizarin

The effect of changing the initial alizarin concentration was investigated by altering the initial concentration from 200 to 2000 mg/L. To investigate the influence of initial concentration of alizarin on its extraction by DES-3, the alizarin solution and DES were combined in a culture tube and mixed for 5 min at room temperature in a $V_{DES}$:$V_W$ ratio of 1:10 mL.

### 3.3.4. Effect of Volume Ratio

The effect of the ratio of the volume of the DES ($V_{DES}$) to the volume of the alizarin solution ($V_w$) is referred to as the effect of volume ratio. During the phase ratio effect, the volume of DES was held constant at 0.15 mL while the volume of the alizarin solution was changed from 1.5 to 9 mL, as in a volumetric ratio changing from 1:10 to 1:60.

### 3.3.5. Effect of Temperature

The temperature of the solution must be adjusted because it plays an essential role in LLE extraction. The temperature was changed in the range of 25–60 °C to investigate the influence of temperature on the extraction of alizarin from a water stream.

### 3.3.6. Effect of Salt

Various amounts of sodium chloride (NaCl) were added to the alizarin solution in concentrations ranging from 2–15% to evaluate the influence of salt on the removal efficiency of alizarin by DES-3. The 200 mg/L alizarin solution containing varying amounts of salt was combined with DES-3 in a 1:10 $V_{DES}$:$V_W$ phase ratio and vortexed for 5 min at room temperature.

### *3.4. Cell Viability Assays*

To determine the effects of DES on human cells, 3-(4, 5-dimethylthiazol-2-yl)-2, 5-diphenyl-2H-tetrazolium bromide (MTT) assays were accomplished as defined previously [30,31]. Briefly, a confluent monolayer of human cells was exposed to DES at various concentrations (i.e., 0.25, 0.5, 1.0, and 2.0 mM) and incubated at 37 °C in the presence of 5% $CO_2$ and 95% humidity overnight. Next, MTT solution (5 mg/mL) was added to serum-free media and added to each well of a 96-well plate. The plate was then incubated at 37 °C for 3 h. After this incubation, 100 μL of filtered sterilized Dimethyl sulfoxide (DMSO) was added to each well. The plate was then wrapped in aluminum foil and kept on an orbital shaker to fully dissolve the crystal formed by the viable cells. For negative control, DMSO alone was used, while positive control was 1-butyl-1-methylpyrrolidinium bis-(triflouromethylsulfonyl) imide a commercial IL. Finally, % cell viability was measured as follows: % Cell viability = (Test sample $_{Mean OD}$/Negative control $_{Mean OD}$) × 100.

## 4. Results and Discussion

### *4.1. Process Optimization*

#### 4.1.1. Screening of the DESs

The alizarin extraction from aqueous phase was assessed and evaluated using the six distinct DESs synthesized by combining various kinds of HBD and HBA. DES-1, DES-2, DES-3, DES-4, DES-5 and DES-6 had an alizarin extraction efficiency of 63.93, 37.18, 98.02, 56.82, 8.07 and 84.32 percent, respectively. DES-3's alizarin extraction efficiency was significantly higher than that of other DESs, as seen in Figure 5. However, all DESs are capable of extracting alizarin from a water stream. The hydrophobicity of the DES increases as the alkyl chain length grows, and clearly TOA has a longer chain length compared to THA. As a result, the hydrophobic DES phase of TOA-based DESs may interact better with alizarin extraction. Therefore, DES-3 was chosen for the optimization study since it had the highest extraction efficiency of all the DESs.

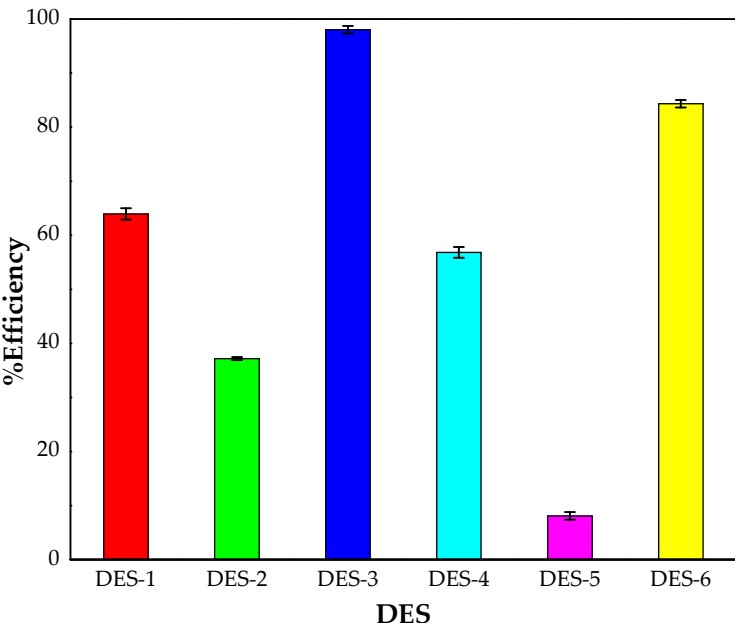

**Figure 5.** Screening of the synthesized DESs for alizarin extraction (initial concentration of alizarin = 200 mg/L, pH = 6.37, contact time = 5 min, DES dosage = 0.15 mL at 25 °C).

### 4.1.2. Effect of pH

Figure 6 shows the effect of pH on alizarin extraction from aqueous solution by DES-3. The extraction efficiency of DES-3 for alizarin increases when the pH of the alizarin solution increases from basic to acidic pH, as shown in Figure 6. The favorable interaction of deprotonated alizarin with positively charged amine-based DES may account for the improved extraction efficiency in lower acidic conditions. Because alizarin's isoelectric point is 4.8, it will operate as a conjugate base at pH levels higher than this, preferring to interact with positive charge carrying DES [32]. At pH 6.37, which is the alizarin solution's natural pH, the maximum removal efficiency of DES-3 for alizarin was found; following that, the removal efficiency remained nearly constant at pH values ranging between 6 and 2.

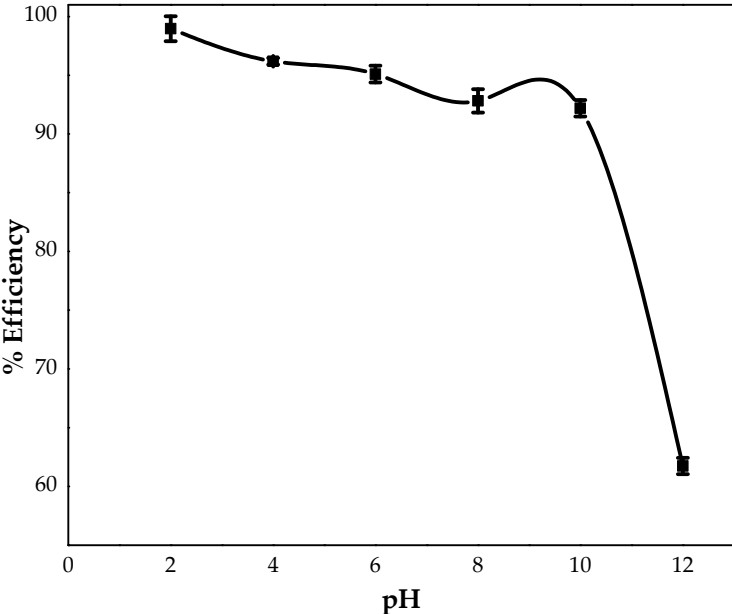

**Figure 6.** The effect of initial pH of alizarin solution on the extraction efficiency by DES-3 (initial concentration of alizarin = 200 mg/L, contact time = 5 min, DES dosage = 0.15 mL at 25 °C).

### 4.1.3. Effect of Time

The influence of vortex mixing duration on alizarin extraction in the LLE process is significant. It also indicates how long it will take to reach the point of equilibrium. This parameter is useful in determining how long it takes the LLE process to reach equilibrium at both laboratory and industrial scales. Figure 7 shows the effect of vortex mixing time on alizarin extraction. Six separate LLE trials were carried out at intervals of 0.5, 1, 3, 4, 5 and 10 min. The results showed that when the contact period is between 0.5 and 1 min, the alizarin extraction is lower, increasing from 95.55 to 98.02 percent as the contact time is increased from 1 to 5 min. Due to the incomplete mixing of DES-3 with alizarin solution at extremely short mixing intervals, such as 0.5 and 1 min, extraction efficiency is slightly lower. However, the considerable increase in extraction effectiveness after 4 min indicates that DES-3 and alizarin have completely mixed. The rate of alizarin extraction is extremely quick, and equilibrium is reached in approximately 5 min. Moreover, experiments were carried out using a higher mixing period, namely 10 min, to confirm the rapid extraction mechanism; however, no improvement in extraction efficiency was detected after 5 min. This demonstrates that the synthesized DES is capable of rapidly extracting alizarin from the aqueous phase. Flieger et al. employed a bi-phase system based on ionic liquids that included dipotassium hydrogen phosphate and 1-butyl-3-methyl-imidazolium chloride to extract alizarin dye from aqueous solution. They investigated the effect of mixing time on extraction efficiency from 1 to 10 min, and after exactly 4.6 min of centrifugation, the maximum extraction was reached [29].

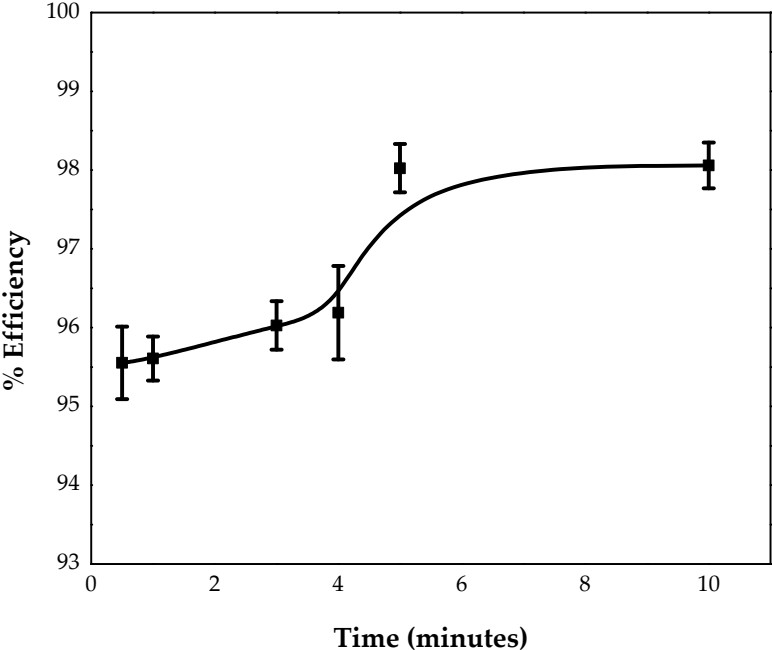

**Figure 7.** Effect of time on alizarin extraction by DES-3 (initial concentration of alizarin = 1000 mg/L, pH = 6.37, DES dosage = 0.15 mL at 25 °C).

### 4.1.4. Effect of Initial Concentration of Alizarin

Different concentrations of alizarin solution, ranging from 200 to 2000 mg/L, were examined to confirm the effects of initial alizarin concentration on the extraction of alizarin by DES-3 from aqueous phase. Variations in dye concentration in aqueous solution and their impact on DES extraction efficiency may aid in the best dye concentration selection for DES treatment. Figure 8 shows the effect of alizarin solution concentration on extraction efficiency. The obtained results demonstrated that when a low alizarin concentration is utilized in LLE, the DES has a higher extraction. DES-3 extraction efficiency for alizarin was 96.19, 95.1, 93.86, 86.32 and 78.91% at 200, 500, 1000, 1500 and 2000 mg/L, respectively.

The experimental results revealed that the extraction efficiency was reduced as the starting concentration of alizarin increased, which is consistent with previous findings [33]. It can be clearly seen that when the initial alizarin dye concentration was increased to 1500 mg/L, there was a significant drop in extraction efficiency (86.32%). This decrease in extraction efficiency may be related to a decrease in the quantity of accessible free DES molecules for alizarin extraction.

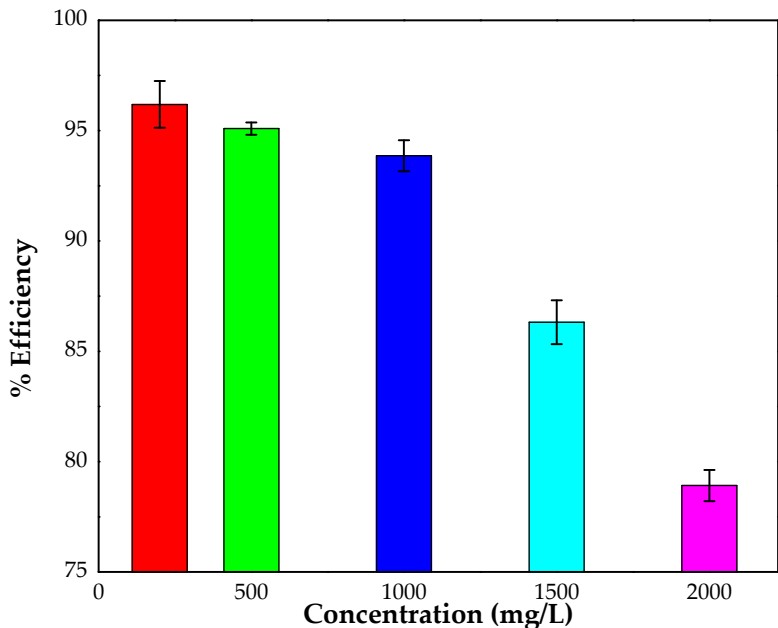

**Figure 8.** Effect of initial alizarin concentration on the extraction efficiency by DES-3 (pH = 6.37, contact time = 5 min, DES dosage = 0.15 mL at 25 °C).

4.1.5. Effect of Volume Ratio

The volume ratio can be used to determine how much DES is needed to extract alizarin from a certain volume and the exact concentration of aqueous alizarin solution. To determine the optimal $V_{DES}:V_w$, a number of experiments were conducted. Figure 9 shows a plot of percentage extraction efficiency versus volumetric ratios of DES to alizarin solution ($V_{DES}:V_w$). The extraction efficiency declined as the $V_{DES}:V_w$ ratio increased, as illustrated in the graph. The extraction efficiency of alizarin from aqueous media utilizing $V_{DES}:V_w$ in the ratios of 1:10, 1:20, 1:30, 1:40 and 1:60 was 98.02, 96.38, 95.73, 91.23 and 88.69 percent, respectively. The results show that DES-3 extracts alizarin from a huge volume of alizarin solution quite efficiently. When the volume of wastewater was raised to 9 mL ($V_{DES}:V_W = 1:60$), the percentage removal reduced marginally, possibly due to saturation of DES-3 with alizarin molecules, resulting in a minor decrease in DES-3′s efficiency. Kakavandi et al. utilized a T-junction microchannel of organic materials for the extraction of alizarin from aqueous phase. The extraction efficiency of alizarin was greatly improved when the DES volume was increased in comparison to the dye aqueous solution volume [33].

4.1.6. Effect of Temperature

The transfer of alizarin from the aqueous to the DES phase is influenced by temperature. The extraction of alizarin using DES-3 was carried out at four different temperatures: 25, 30, 40, and 60 °C, to emphasize the temperature effect on alizarin extraction. Figure 10 shows the effect of temperature on alizarin extraction by the synthesized DES-3. Temperature had no effect on the extraction efficiency of alizarin, according to the graph. A decrease of around only 3% in the extraction efficiency was noticed when the temperature was raised to 60 °C. This minor decrease in extraction efficiency may be attributed to the decrease in

the interaction between the DES and alizarin at higher temperature, indicating that the enthalpy of solubilization of alizarin in DES is slightly exothermic. The expected result of this effect is the facilitated backward mass transfer of alizarin into the aqueous phase, hence lowering its extraction efficiency. The findings are consistent with those found in the literature [34].

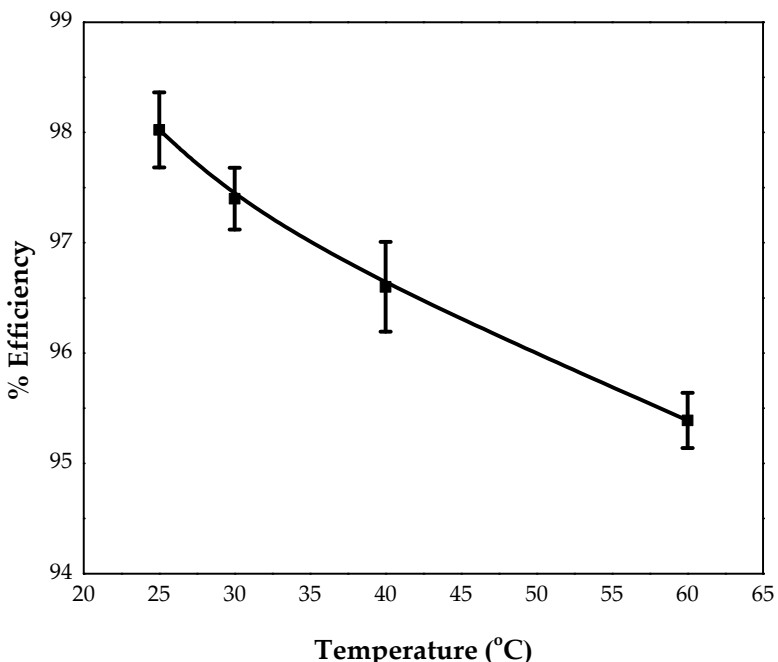

**Figure 9.** Effect of DES to alizarin volume ratio on the extraction efficiency (initial concentration of alizarin = 1000 mg/L, pH = 6.37, contact time = 5 min, DES dosage = 0.15 mL at 25 °C).

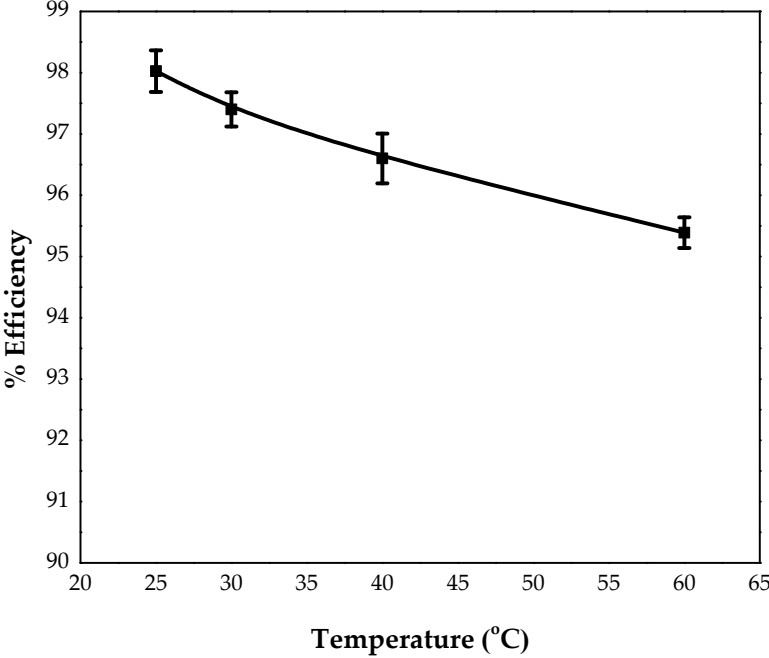

**Figure 10.** Effect of temperature on alizarin extraction by DES-3 (initial concentration of alizarin = 1000 mg/L, pH = 6.37, contact time = 5 min and DES dosage = 0.15 mL).

4.1.7. Effect of Salt (Ionic Strength of Alizarin Solution)

To mimic the natural conditions of wastewater, the extraction efficiency of DES-3 for alizarin aqueous solution with varied salt concentrations was investigated. The effect of salt on the extraction efficiency of DES-3 for alizarin is seen in Figure 11. When 2 percent NaCl is added to alizarin solution, the extraction efficiency drops from 99.33 percent to 96.9%. When the salt content in the aqueous alizarin solution was increased to 15 percent, the extraction efficiency increased to 98.2 percent. The salting-out effect, in which alizarin molecules are driven to migrate from the aqueous to the DES phase, may explain the overall drop in extraction efficiency. The use of DES to improve dye extraction efficiency has already been documented elsewhere in the literature [35].

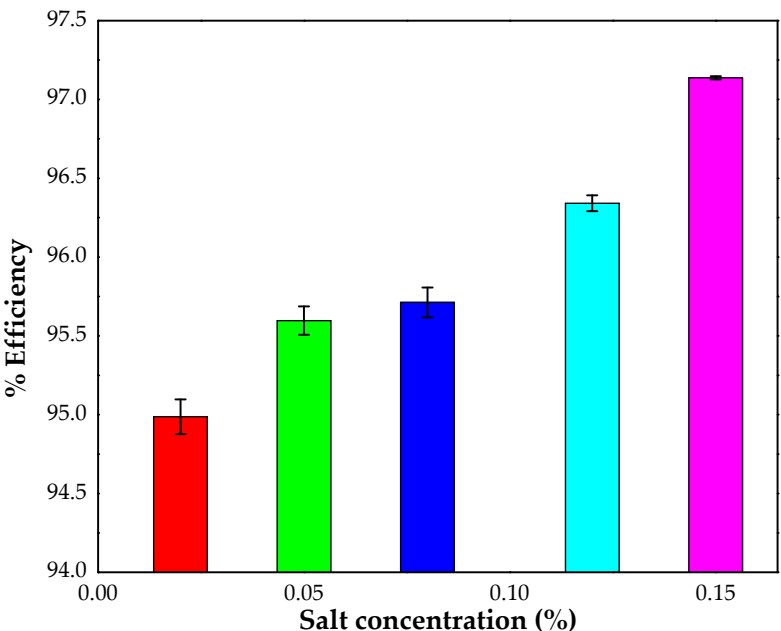

**Figure 11.** Effect of salt concentration on alizarin extraction by DES-3 (pH of 6.37, 5 min, $C_i$ = 1000 mg/L, DES dosage = 0.15 mL and at 25 °C).

4.1.8. Thermodynamic Analysis

Thermodynamic parameters including enthalpy ($\Delta H$), Gibbs free energy ($\Delta G$), and entropy ($\Delta S$) were determined in a temperature range of 25 to 60 °C to better understand the impact of heat on the extraction of alizarin from aqueous solution utilizing DES-3. These variables will aid in determining the nature of the reaction, namely whether it is viable and spontaneous. The enthalpy of the reaction at specified temperatures was calculated using the Van't Hoff Equation (3). A graph was drawn between ln(K) and 1/T in order to compute the enthalpy. To fit the data points, a third-order polynomial was applied. At the stated temperatures, the slopes of this polynomial were calculated, yielding $\Delta H$ values. Then, using Equation (5), $\Delta G$ was computed and finally, using Equation (6), $\Delta S$ was also determined.

$$\frac{d(\ln D)}{d(1/T)} = -\frac{\Delta H}{R} \tag{3}$$

$$D = \frac{(C_i - C_f)}{Cf} \tag{4}$$

$$\Delta G = -RT \ln K \tag{5}$$

$$\Delta S = \frac{\Delta H - \Delta G}{T} \tag{6}$$

where R is the ideal gas constant and T is the temperature in kelvin.

Table 3 lists the values of all thermodynamic parameters. At all temperatures, negative values for ΔG confirm the spontaneity of the alizarin dye extraction by DES-3 process. Increasing temperature resulted in a decrease in ΔG values, indicating that the reaction was less spontaneous. For alizarin extraction, ΔH values are negative (exothermic nature), indicating that the interaction of alizarin with DES-3 is advantageous at low temperatures. At 25 °C, entropy is positive, implying that entropy is also driving alizarin extraction by DES-3; therefore, at this temperature, the hydrophobic interaction is confirmed.

**Table 3.** Thermodynamic parameters for alizarin extraction in an aqueous media utilizing DES-3.

| Temperature (K) | ΔG (kJ/mol) | ΔH (kJ/mol) | ΔS (kJ/mol) |
|---|---|---|---|
| 298.15 | −9.67 | −0.32 | 0.031 |
| 303.15 | −9.13 | −0.89 | 0.027 |
| 313.15 | −8.17 | −14.66 | −0.02 |
| 333.15 | −8.39 | −82.31 | −0.22 |

### 4.2. Solubility of DES-3 in Water

The DES to be used for liquid–liquid extraction should be hydrophilic with no detectable solubility in aqueous solution. To test this restriction for DES-3, an experiment was performed by recording the UV-visible spectra for pure water, pure DES-3 and water after mixing with DES-3 (which was vortexed for 5 min and centrifuged for 10 min). A syringe was inserted inside the centrifuge tube to withdraw some of the water and record its spectra. Figure 12 displays the results. Inspection of this figure reveals that water mixed with DES-3 after separation is completely free from any trace of DES-3. Hence, it be concluded that DES-3 is highly hydrophobic and does not cause any contamination of water.

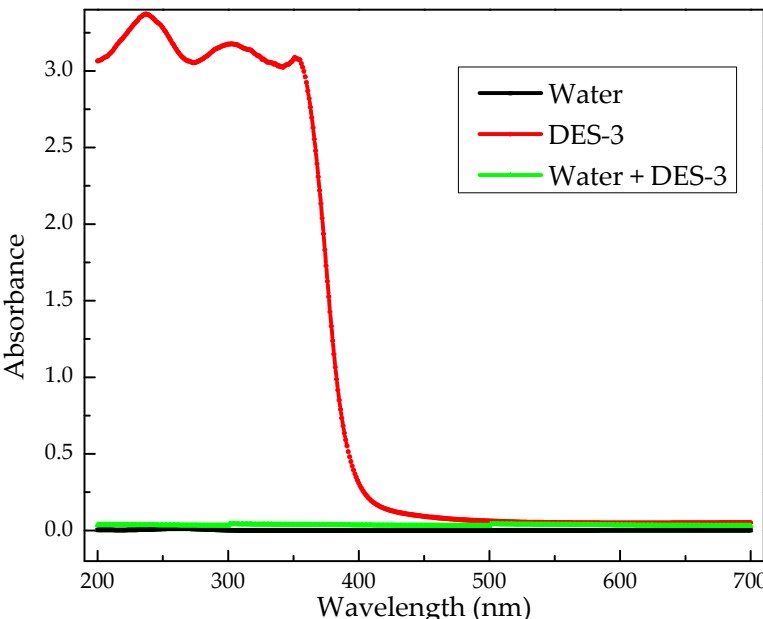

**Figure 12.** UV-Vis spectra for pure water (black), pure DES-3 (red) and water after mixing with DES-3 and separation (green).

### 4.3. Thermogravimetric Analysis (TGA) of DES-3

TGA was used to investigate the thermal stability of DES-3 in a nitrogen environment. The results are shown in Figure 13. Inspection of this figure reveals that the onset decomposition temperature of DES-3 was 320 °C. Furthermore, the figure shows that at 400 °C, DES-3 was completely disintegrated. These results are aligned with the reported values in the literature for similar systems [36,37].

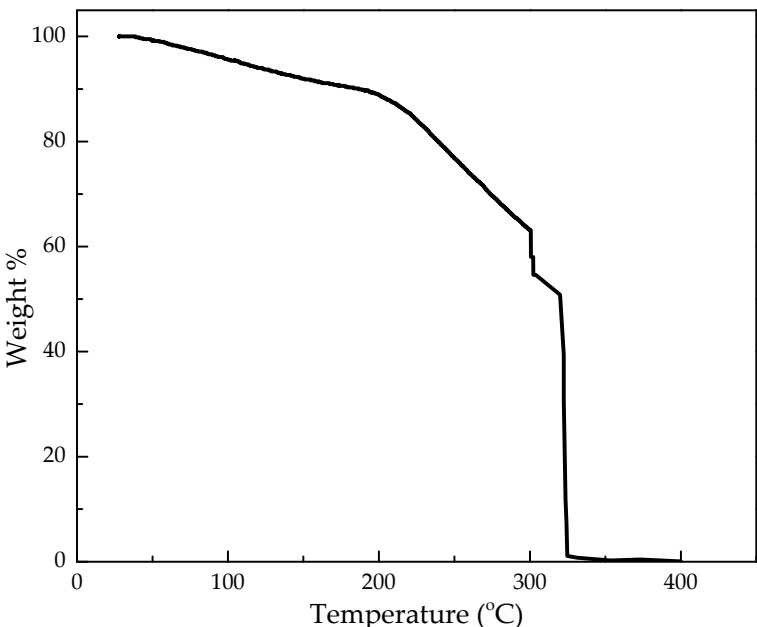

**Figure 13.** TGA profile of DES-3.

*4.4. FTIR Analysis*

FTIR spectra were recorded for all samples in the range from 400 to 4000 cm$^{-1}$, as shown in Figure 14. The peaks at 2931 and 2791 cm$^{-1}$ represent the stretching vibration of the CH$_2$ group and methylene in TOA, respectively [38]. The presence of the frequency at 3435 cm$^{-1}$ is due to the vibration of N-H in the TOA sample. From the spectrum, it was detected that a peak appearing at 1470 cm$^{-1}$ (indicating N-H$_2$ symmetric stretching vibration) in the case of pure TOA shifted to 1460 cm$^{-1}$ in DES and then to 1450 cm$^{-1}$ for alizarin-loaded DES. A characteristic peak shown at 1290 cm$^{-1}$ resulted in a C–N stretching vibration. A peak of pure TOA at 1094 cm$^{-1}$ (C–N stretching vibrations) shifted to 1030 cm$^{-1}$ after formation of DES with salicylic acid and to position 1024 cm$^{-1}$ when loaded with dye [39,40].

*4.5. Cell Viability Assays*

The overall results from MTT assays revealed that the all the DES showed minimal to moderate cytotoxicity against human cells (Figure 15). DES-1 and DES-2 showed negligible cytotoxic effects at all concentrations against human cells. DES-3, having maximum alizarin extraction efficiency, showed minimal cytotoxicity at the first three concentrations, whereas DES-3 at 2.0 mM had weak cytotoxic activity, showing 35% cell death. Similarly, DES-4 and DES-5 had similar trends in cytotoxicity, showing weak cytotoxicity only at 1.0 and 2.0 mM concentrations. Finally, DES-6 exhibited minimal cytotoxic effects at 0.125 and 0.250 mM, showed weak cytotoxicity at 0.50 and 1 mM, and showed moderate to high cell death at 2.0 mM. Thus, our data show that our manufactured DES had less harmful effects than the commercially available IL, i.e., 1-butyl-1-methylpyrrolidinium bis-(trifluoromethylsulfonyl) imide [41]. Our findings are consistent with previous research that tested several ILs against human cells. For example, in a recent study, DES based on hydrogen bond donors and hydrogen bond acceptors showed minimal cytotoxic effects against human cells [42]. On the contrary, in another study it was discovered that the [Chol]Cl-based DES examined are relatively cytotoxic to all cell lines [43]. Overall, our cell viability assay results revealed that our lab-made DESs have more cell viability impacts than commercial ILs.

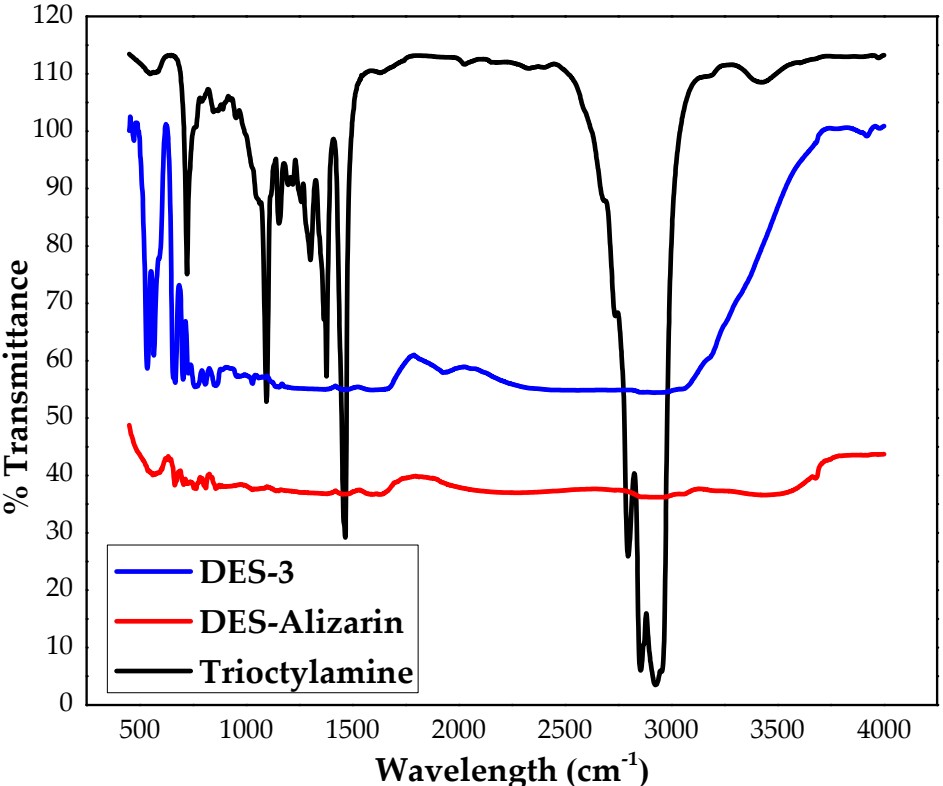

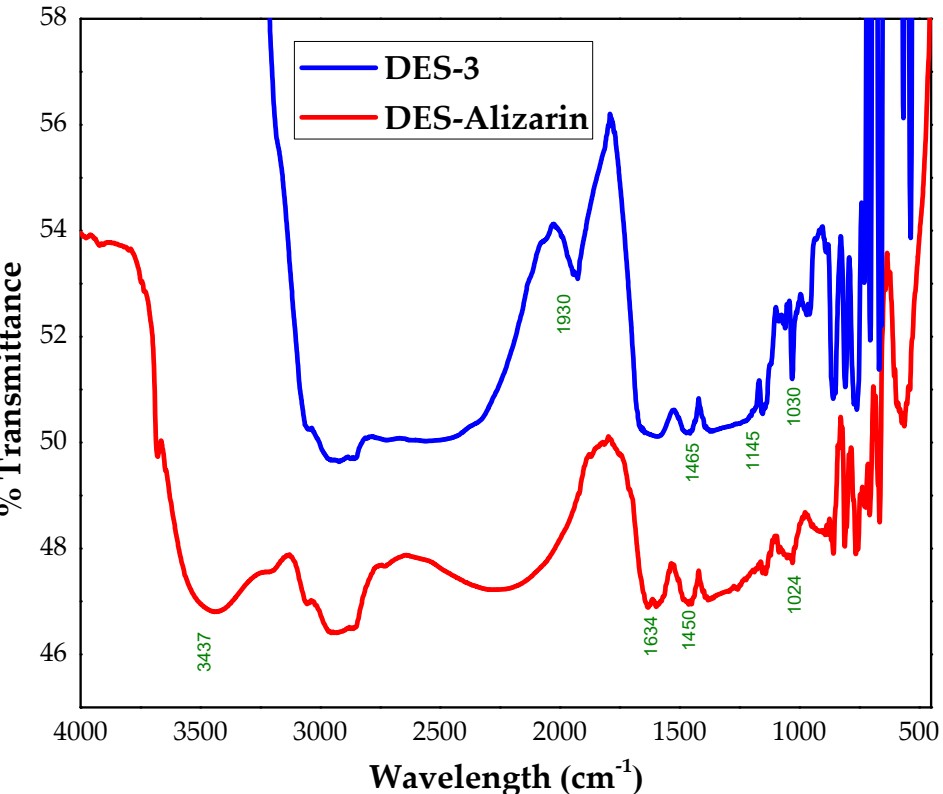

**Figure 14.** FTIR spectra for trioctylamine and DES-3 before and after alizarin extraction.

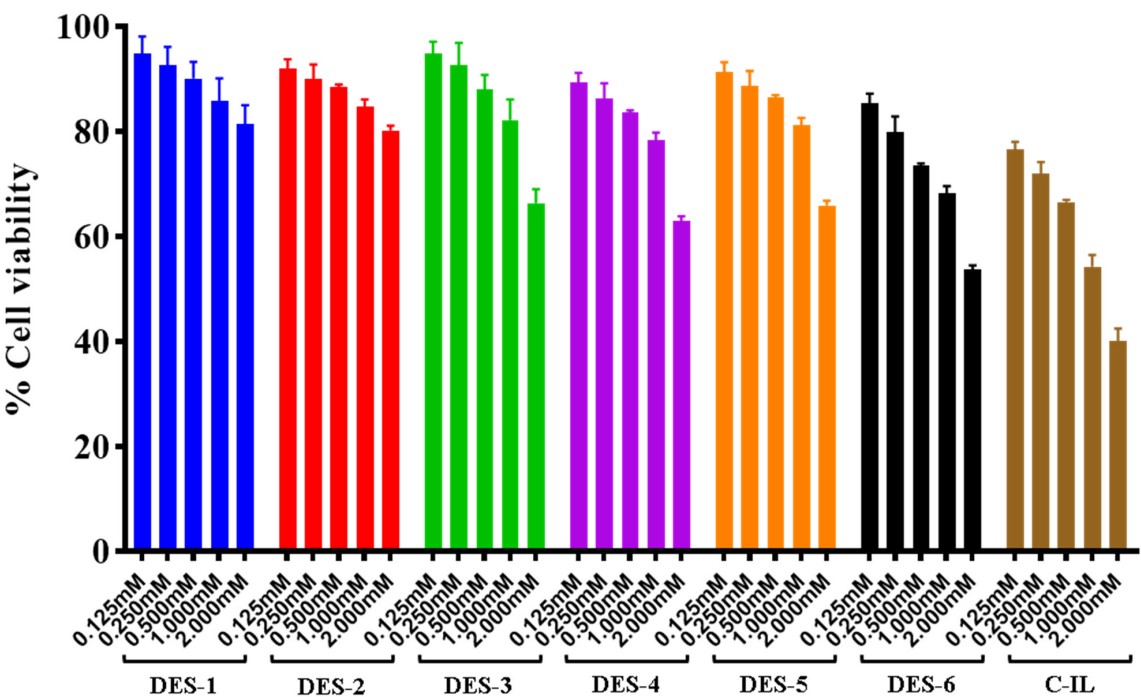

**Figure 15.** Effect of deep eutectic solvents on human cell line at their graduate concentrations.

## 5. Comparative Study with Literature

The optimum alizarin extraction efficiency of DES-3 was compared to that of other extractors previously reported in the literature. Table 4 shows literature values of the extraction efficiency of other dyes using different solvents. Inspection of Table 4 reveals that DES-3 is among the best solvents for extraction of dyes from aqueous solution.

**Table 4.** Literature reports on the extraction of different dyes using various solvents through LLE.

| Extraction Solvent | Target Analyte | %E | Reference |
|---|---|---|---|
| Tricaprylmethylammonium thiocyanate | Methyl orange | 89.09 | [44] |
|  | Methylene blue | 64.14 |  |
| Tetrabutyl ammonium bromide | Anionic dyes | 98 | [45] |
| *N*-butyl, *N*-methyl pyrrolidinium bis(trifluoromethanesulfonyl) amide | Navy 5RE (acid blue) | 98 | [10] |
|  | Acid red | 98 |  |
| Phosphonium based ILs | Chloranilic acid | ~100 | [46] |
|  | Indigo blue |  |  |
|  | Sudan III |  |  |
| Acetonitrile | Methylene blue | 60 | [47] |
|  | Sunset yellow | 70 |  |
| Hexafluoroisopropanol (HFIP)-based DESs | Methylene blue |  | [35] |
|  | Tartrazine |  |  |
|  | Sudan III |  |  |
| Salicylic acid | Methyl violet | 96 | [48] |
| Lycolic acid and choline chloride based DESs | Eriochrome black T | 90 | [17] |
| Butyl-methyl imidazolium chloride and potassium dihydrogen phosphate | Alizarin Red S | 95 | [29] |
| DES-3 | Alizarin | 98.02 | This work |

## 6. Conclusions

Six distinct types of hydrophobic DESs were synthesized and tested for alizarin aqueous phase extraction, employing three different HBAs (trioctylphosphine (TOP), trioctylamine (TOA) and trihexylamine (THA) and two different HBDs (salicylic acid (SA) and malonic acid (MA). Although all six DESs removed some alizarin from aqueous solution, the extraction efficiency of DES-3 was exceptional, reaching 98.02%. As a result, DES-3 was chosen, and all of the alizarin extraction parameters were adjusted accordingly. At 25 °C, the extraction reaction reached equilibrium in almost 5 min. With increasing initial alizarin content and $V_{DES}:V_W$, the extraction efficiency of alizarin solution declined. Thermodynamic analysis showed that the alizarin extraction reaction is spontaneous and exothermic, and hence extraction efficiency decreases as temperature rises. The findings of the toxicity tests proved that DES-3 had a low level of cytotoxicity. Overall, this research proved that DES-3 has the ability to remove alizarin contamination from the aqueous phase.

**Author Contributions:** Conceptualization, A.S.K. and T.H.I.; methodology, T.H.I.; software, validation, M.K. and T.H.I.; formal analysis, N.Y. and M.F.H.; investigation, N.Y., N.A. and R.S.; resources, T.H.I. and M.K.; data curation, writing—original draft preparation, N.Y. and A.S.K.; writing—review and editing, T.H.I., M.K. and P.N.; visualization, N.A.K.; supervision, T.H.I. and M.K.; project administration, T.H.I. and M.K.; funding acquisition, T.H.I. All authors have read and agreed to the published version of the manuscript.

**Funding:** This research received no external funding.

**Institutional Review Board Statement:** Not applicable.

**Informed Consent Statement:** Not applicable.

**Data Availability Statement:** Not applicable. All generated data are included in the manuscript.

**Conflicts of Interest:** The authors declare no conflict of interest.

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
