# Peer review of "Amine-Based Deep Eutectic Solvents for Alizarin Extraction from Aqueous Media"

_processes, doi:10.3390/pr10040794_

Round 1
Reviewer 1 Report
It is an interesting article on Alizarin Extraction from Aqueous Media.
According to the results analysis, in my opinion, the best conditions for liquid liquid extraction under the studied conditions are presented, but those are not strictly the optimal conditions. To obtain the optimal conditions, a mathematical or statistical analysis would be required, (Abstract, 1. Introduction, 3.3. Single parameter optimization process, 4.1. Process optimization).
Improve Figure 10 and the equations: 1, 2, 3, 4, 5.
It is recommended to add a graphical abstract.
Author Response
Review 1
Comments and Suggestions for Authors
It is an interesting article on Alizarin Extraction from Aqueous Media.
We would like to the thank the reviewer for his nice comments and encouragement
According to the results analysis, in my opinion, the best conditions for liquid-liquid extraction under the studied conditions are presented, but those are not strictly the optimal conditions. To obtain the optimal conditions, a mathematical or statistical analysis would be required (Abstract, 1. Introduction, 3.3. Single parameter optimization process, 4.1. Process optimization).
We agree with the reviewer’s comment. However, in this work, we tried to optimize the method with respect to process variables such as pH, contact time, dye initial concentration, volume ratio, temperature and salt. Each variable was optimized with statistical analysis. The standard deviation for each parameter is now added to each of the figures (Figs 5-11).
Improve Figure 10 and the equations: 1, 2, 3, 4, 5.
We have improved figure 10, modified the caption of all other figures and improved equations 1-5.
It is recommended to add a graphical abstract.
We would to thank the reviewer for his suggestion. Now, graphical abstract is added to the manuscript.
Reviewer 2 Report
In the manuscript, Ibrahim et al. proposed a new method to extract alizarin from aqueous solutions using amine-based deep eutectic solvents (DESs). The DESs show high efficiency to remove alizarin from aqueous solutions. The manuscript tis well organized. The reviewer suggests it can be accepted after addressing the following questions or problems.
- In Figure 3, the third structure is tri-n-heptylamine, but authors call it trihexylamine.
- Salicyclic acid and malonic acid are easily dissolved in water especially in water. If the DESs are dissolved in eater, they may pollute water. The solubility of DESs in aqueous solutions should be reported.
- How to regenerate the DESs after extraction? Cycle data of extraction is suggested to report.
- The authors state that “Only around 3% decrease in the extraction efficiency was noticed when the temperature was raised to 60 oC. This minor decrease in extraction efficiency could be attributed to the DES’s viscosity decreasing as the temperature rises”. The explanation is not right.
Author Response
Review 2
Comments and Suggestions for Authors
In the manuscript, Ibrahim et al. proposed a new method to extract alizarin from aqueous solutions using amine-based deep eutectic solvents (DESs). The DESs show high efficiency to remove alizarin from aqueous solutions. The manuscript tis well organized. The reviewer suggests it can be accepted after addressing the following questions or problems.
We would like to thank the reviewer for his sharp summary and encouragement.
In Figure 3, the third structure is tri-n-heptylamine, but authors call it trihexylamine.
We thank the reviewer for his thorough comment. This typo error is now fixed (Figure 4).
Salicyclic acid and malonic acid are easily dissolved in water especially in water. If the DESs are dissolved in eater, they may pollute water. The solubility of DESs in aqueous solutions should be reported.
We agree with the reviewer comment. An experiment to confirm the hydrophobicity of DES-3 was performed in which the UV-visible spectra was recorded for pure water, pure DES-3 and water after mixing with DES-, vortexed for 5 min and separated by centrifugation for 10 minutes. The results indicated that DES-3 didn’t dissolve in water rending it as highly hydrophobic. The results are added to the manuscript in section 6.2 (page 18 & 19).
How to regenerate the DESs after extraction? Cycle data of extraction is suggested to report.
We acknowledge the importance of question raised by the reviewer. Up to this point, in the literature, it was not mentioned how to regenerate DESs when utilized in liquid-liquid extraction processes. However, for gas absorption, DES were regenerated through air bubbling. Hence, this point required further investigation for liquid and solid materials hosted in DES. Some suggestion that will be investigated in future work might include selected advance oxidation processes, controlled thermal decomposition processes and ion exchange processes.
The authors state that “Only around 3% decrease in the extraction efficiency was noticed when the temperature was raised to 60 oC. This minor decrease in extraction efficiency could be attributed to the DES’s viscosity decreasing as the temperature rises”. The explanation is not right.
We agree with the reviewer’s comment and hence, this sentence was corrected in the manuscript to indicate that this slight decrease in efficiency could be attributed to decrease in the interaction between DES and alizarin with increasing temperature (page 16).
Reviewer 3 Report
The paper entitled 'Amine-Based Deep Eutectic Solvents for Alizarin Extraction from Aqueous Media' describes the peparation of hydrophobic DES for the extraction of Alizarin. Unfortunately, the work is based on incorrect assumptions made by the authors, which makes it unsuitable for publication.
In the first instance, the authors confuse ionic liquids and DES. DES by their definition are eutectic mixtures that show a marked deviation from the ideal eutectic (e.g. 10.1002/cssc.202001331). The authors refer to new DES without verifying their real "deep" nature. In addition, mixtures made from acid and amine in a 1:1 ratio are to all intents and purposes Protic Ionic Liquids (PILs). The literature on PILs is extensive and explains well how they are derived from the reaction of an acid and a base with a certain pH difference between the two species. For phosphines, which are not mentioned in the title of the paper, the problem is even greater. They are highly sensitive to oxygen and easily give rise to phosphinoxides. In addition, in the presence of strong acids, they too can be regarded as PILs, but with a very low oxidation stability.
In addition, the toxicity of long-chain amines and phosphines (including their oxides) is marked, making them unsuitable systems from the point of view of eco-sustainability.
For these reasons, the work is completely unsuitable for publication.
Author Response
Review 3
Comments and Suggestions for Authors
The paper entitled 'Amine-Based Deep Eutectic Solvents for Alizarin Extraction from Aqueous Media' describes the preparation of hydrophobic DES for the extraction of Alizarin. Unfortunately, the work is based on incorrect assumptions made by the authors, which makes it unsuitable for publication.
In the first instance, the authors confuse ionic liquids and DES. DES by their definition are eutectic mixtures that show a marked deviation from the ideal eutectic (e.g. 10.1002/cssc.202001331). The authors refer to new DES without verifying their real "deep" nature. In addition, mixtures made from acid and amine in a 1:1 ratio are to all intents and purposes Protic Ionic Liquids (PILs). The literature on PILs is extensive and explains well how they are derived from the reaction of an acid and a base with a certain pH difference between the two species. For phosphines, which are not mentioned in the title of the paper, the problem is even greater. They are highly sensitive to oxygen and easily give rise to phosphinoxides. In addition, in the presence of strong acids, they too can be regarded as PILs, but with a very low oxidation stability.
Allow us to disagree with the reviewer’s comment. We have mentioned in the introduction that DESs are considered as modified IL with additional advantages. Hence, we didn’t mix between the two and accordingly, we applied DES solvent to alizarin extraction. In addition, phosphine based DES (DES-1 and DES-2) were also prepared and their potentials for alizarin extraction were reported. However, they weren’t as effective as amine based (DES-3) extractor. Furthermore, these types of HBA and HBD DESs systems were reported in the literature (G. Almustafa, R. Sulaiman, M. Kumar, I. Adeyemi, H. A. Arafat, and I. AlNashef, “Boron extraction from aqueous medium using novel hydrophobic deep eutectic solvents,” Chem. Eng. J., vol. 395, p. 125173, 2020, doi: https://doi.org/10.1016/j.cej.2020.125173.; Y. Shi, D. Xiong, Y. Zhao, T. Li, K. Zhang, and J. Fan, “Highly efficient extraction/separation of Cr (VI) by a new family of hydrophobic deep eutectic solvents,” Chemosphere, vol. 241, p. 125082, 2020, doi: https://doi.org/10.1016/j.chemosphere.2019.125082.). In this work, we followed the methodology outlined in these references for the preparation of the reported DESs.
In addition, the toxicity of long-chain amines and phosphines (including their oxides) is marked, making them unsuitable systems from the point of view of eco-sustainability.
We have performed toxicity experiments and the results are already reported in the manuscript. The conclusion is “DES-6 exhibited minimal cytotoxic effects at 0.125 mM”.
For these reasons, the work is completely unsuitable for publication.
We regret the reviewer’s recommendation and we hope that he will reconsider after our detilaed responses to his comments
Reviewer 4 Report
Yasir et al prepared six hydrophobic DESs for high efficiency extract of Alizarin from aqueous media. It can be accepted for publication. However, the following concerns are suggested to be addressed before the acceptation.
I do not see the proof of the hydrophobicity of the DESs.
Are these DESs thermal stable? What are the decomposition temperature of them?
It is preferable to provide a comparison of this work with previous reported materials.
There are some examples on hydrophobic DESs in a recently published review Chem Soc Rev, 2021, 50, 8596-8638.

Author Response
Review 4
Comments and Suggestions for Authors
Yasir et al prepared six hydrophobic DESs for high efficiency extract of Alizarin from aqueous media. It can be accepted for publication. However, the following concerns are suggested to be addressed before the acceptation.
I do not see the proof of the hydrophobicity of the DESs.
To address the reviewer’s comment and proof the hydrophobicity of DES-3 an experiment was performed in which the UV-visible spectra was recorded for pure water, pure DES-3 and water after mixing with DES-, vortexed for 5 min and separated by centrifugation for 10 minutes. The results indicated that DES-3 didn’t dissolve in water rending it as highly hydrophobic. The results are added to the manuscript in section 6.2 (page 18 & 19).
Are these DESs thermal stable? What are the decomposition temperature of them?
To address the reviewer’s comment, TGA analysis was conducted to confirm the thermal stability of DES-3. It was found that the decomposition of DES-3 start to occur at 320 oC and it completely disintegrate at 400 oC. The results are added to the manuscript in section 6.3 (page 19 & 20) and a sentence was also added in the abstract to summarize the results.
It is preferable to provide a comparison of this work with previous reported materials.
We thank the reviewer for his valuable recommendation. Comparative study was added to the manuscript for different solvents in extracting dyes and their respective removal efficiency. (Section 7, Page 23)
Round 2
Reviewer 2 Report
The reviewer thinks that the authors have responded reasonably and revised their manuscript according to the comments. The reviewer suggests it can be accepted.
Author Response
Thank you very much for reviewing the manuscript for us.
All the best to your work and I wish you a nice day.
Reviewer 3 Report
I thank the authors for their reply. I probably failed to make it clear. It was not my purpose to say that the difference between DES and ILs was not explained. However, the definition "DESs, which are essentially modified ILs" is fundamentally wrong. In fact, DESs are a mixture of two compounds, even non-ionic compounds, which have a deviation from the ideal eutectic. In this regard, I report one of the works of Coutinho (considered the father of DES) where he clearly explains what a DES is (10.1007/s10953-018-0793-1). Instead, those reported by the authors are Protic Ionic Liquids (PILs) resulting from the reaction of acid and a Bronsted base in stoichiometric ratio (1:1 in moles). This class of ionic liquids, well known in the literature (10.1021/acs.chemrev.5b00158), differs from DES in that it involves an acid-base reaction leading to two ionic species, one cationic and one anionic. To confirm this, the authors report a high thermal stability typical of PILs, which depends on the difference in pKa of the two precursors (10.1039/C2CP00007E).
To further confirm this, in the work proposed by the authors (G. Almustafa, R. Sulaiman, M. Kumar, I. Adeyemi, H. A. Arafat, and I. AlNashef, "Boron extraction from aqueous medium using novel hydrophobic deep eutectic solvents," Chem. Eng. J., vol. 395, p. 125173, 2020, doi: https://doi.org/10.1016/j.cej.2020.125173.; Y. Shi, D. Xiong, Y. Zhao, T. Li, K. Zhang, and J. Fan, "Highly efficient extraction/separation of Cr (VI) by a new family of hydrophobic deep eutectic solvents," Chemosphere, vol. 241, p. 125082, 2020, doi: https://doi.org/10.1016/j.chemosphere.2019 .125082.) reporting the use of hydrophobic DES in extractive processes do not involve the HBAs and HBDs used by the authors.
For example, in the work reported by Almustafa et al., the use of menthol and thymol with decanol and 2-methyl-2,4-pentanedium are quite different from those reported by the authors, as there is no reaction between the two constituents and no formation of ionic species.
Instead, the work reported by Shi et al. uses tri-octylmethylammonium chloride as HBA, which unlike the amines used by the authors is an ammonium salt where the chloride functions as HBA and not the entire salt.
This misunderstanding happens very often and is due to the recent explosion of the DES field. However, the recent literature on this subject is very clear, detailed and abundant and it is no more possible to publish work that continues to generate confusion and misunderstanding as has happened in the past.
For these reasons, although the results are interesting, the work cannot be accepted in this form. In order to reconsider the work, I suggest removing any reference to DES and instead talk about PILs as this is what it is all about. In addition to this, a new introduction on PILs must be made. Only in this way can the work be reconsidered and evaluated properly.
Author Response
Ms. Jennie Zhu M.Sc
Section Managing Editor
Processes
Re: Manuscript Number: processes-1642668 
 
Reviewer 3
i thank the authors for their reply.
You are most welcome
I probably failed to make it clear. It was not my purpose to say that the difference between DES and ILs was not explained. However, the definition "DESs, which are essentially modified ILs" is fundamentally wrong. In fact, DESs are a mixture of two compounds, even non-ionic compounds, which have a deviation from the ideal eutectic. In this regard, I report one of the works of Coutinho (considered the father of DES) where he clearly explains what a DES is (10.1007/s10953-018-0793-1). Instead, those reported by the authors are Protic Ionic Liquids (PILs) resulting from the reaction of acid and a Bronsted base in stoichiometric ratio (1:1 in moles). This class of ionic liquids, well known in the literature (10.1021/acs.chemrev.5b00158), differs from DES in that it involves an acid-base reaction leading to two ionic species, one cationic and one anionic. To confirm this, the authors report a high thermal stability typical of PILs, which depends on the difference in pKa of the two precursors (10.1039/C2CP00007E).
We agree with reviewer’s comment that there is an analogy between DES and PIL. However, recent reviews on the subject have referred to these materials as DES or ionic liquid (IL) analogues. In addition, thermodynamic studies on few examples of them reveals their DES properties (for example, see Molecules 2022, 27, 1368. https://doi.org/10.3390/molecules27041368). Hence, we believe that the name of DES is widely accepted as appropriate for such systems.
To further confirm this, in the work proposed by the authors (G. Almustafa, R. Sulaiman, M. Kumar, I. Adeyemi, H. A. Arafat, and I. AlNashef, "Boron extraction from aqueous medium using novel hydrophobic deep eutectic solvents," Chem. Eng. J., vol. 395, p. 125173, 2020, doi: https://doi.org/10.1016/j.cej.2020.125173.; Y. Shi, D. Xiong, Y. Zhao, T. Li, K. Zhang, and J. Fan, "Highly efficient extraction/separation of Cr (VI) by a new family of hydrophobic deep eutectic solvents," Chemosphere, vol. 241, p. 125082, 2020, doi: https://doi.org/10.1016/j.chemosphere.2019 .125082.) reporting the use of hydrophobic DES in extractive processes do not involve the HBAs and HBDs used by the authors.
For example, in the work reported by Almustafa et al., the use of menthol and thymol with decanol and 2-methyl-2,4-pentanedium are quite different from those reported by the authors, as there is no reaction between the two constituents and no formation of ionic species.
Yes, we agree with the reviewer that our DES systems are different than those reported in the above mentioned references, however, they have the HBD and HBA properties that are common to all systems used. Furthermore, the stating materials in our systems are not ionic and may be highly bridged by hydrogen bond between the two components, which is markedly different than starting with ionic components that form IL.
Instead, the work reported by Shi et al. uses tri-octylmethylammonium chloride as HBA, which unlike the amines used by the authors is an ammonium salt where the chloride functions as HBA and not the entire salt.
This misunderstanding happens very often and is due to the recent explosion of the DES field. However, the recent literature on this subject is very clear, detailed and abundant and it is no more possible to publish work that continues to generate confusion and misunderstanding as has happened in the past.
For these reasons, although the results are interesting, the work cannot be accepted in this form. In order to reconsider the work, I suggest removing any reference to DES and instead talk about PILs as this is what it is all about. In addition to this, a new introduction on PILs must be made. Only in this way can the work be reconsidered and evaluated properly.
New literature studies in the field are referring to these compounds as DES with emphases on being also IL analogue (Molecules 2022, 27, 1368. https://doi.org/10.3390/molecules27041368). Therefore, we do not agree with the reviewer on this point. There are varying definitions of DES and there is not yet a single universally accepted definition. Therefore, we suggest that the academic editor of the journal should make the final decision on this.
We hope that our manuscript is now acceptable for publication in your Journal in its current form.
Best regards,
Taleb Ibrahim, Ph.D
Corresponding author